# Feature-Decision Level Collaborative Fusion Network for Hyperspectral and LiDAR Classification

**Shenfu Zhang** [1], **Xiangchao Meng** [1,*], **Qiang Liu** [1], **Gang Yang** [2] and **Weiwei Sun** [2]

1 Faculty of Electrical Engineering and Computer Science, Ningbo University, Ningbo 315211, China; 2211100073@nbu.edu.cn (S.Z.); 2001100032@nbu.edu.cn (Q.L.)
2 Department of Geography and Spatial Information Techniques, Ningbo University, Ningbo 315211, China; yanggang@nbu.edu.cn (G.Y.); sunweiwei@nbu.edu.cn (W.S.)
* Correspondence: mengxiangchao@nbu.edu.cn

**Abstract:** The fusion-based classification of hyperspectral (HS) and light detection and ranging (LiDAR) images has become a prominent research topic, as their complementary information can effectively improve classification performance. The current methods encompass pixel-, feature- and decision-level fusion. Among them, feature- and decision-level fusion have emerged as the mainstream approaches. Collaborative fusion of these two levels can enhance classification accuracy. Although various methods have been proposed, some shortcomings still exist. On one hand, current methods ignore the shared advanced features between HS and LiDAR images, impeding the integration of multimodal features and thereby limiting the classification performance. On the other hand, the existing methods face difficulties in achieving a balance between feature- and decision-level contributions, or they simply overlook the significance of one level and fail to utilize it effectively. In this paper, we propose a novel feature-decision level collaborative fusion network (FDCFNet) for hyperspectral and LiDAR classification to alleviate these problems. Specifically, a multilevel interactive fusion module is proposed to indirectly connect hyperspectral and LiDAR flows to refine the spectral-elevation information. Moreover, the fusion features of the intermediate branch can further enhance the shared-complementary information of hyperspectral and LiDAR to reduce the modality differences. In addition, a dynamic weight selection strategy is meticulously designed to adaptively assign weight to the output of three branches at the decision level. Experiments on three public benchmark datasets demonstrate the effectiveness of the proposed methods.

**Keywords:** hyperspectral (HS); light detection and ranging (LiDAR); feature fusion; decision fusion; remote sensing classification

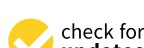



## 1. Introduction

Remote sensing imagery as a means of acquiring ground object information [1] has found extensive applications in geological exploration [2], marine monitoring [3] and disaster management [4]. In recent years, the advancement of sensor technology has facilitated the provision of a diverse range of remote sensing images. Different sensors can provide varying information for land-cover objects within the same geographic area. Hyperspectral data have been widely used in the fine classification of land-cover and land-use due to rich spectral-spatial information. However, it is always difficult to accurately classify ground objects only with hyperspectral data because of the phenomenon of the "same spectrum of foreign objects" [5]. LiDAR data provide high-precision three-dimensional spatial information, which is complementary to hyperspectral data. For example, hyperspectral data fail to distinguish the roads and buildings because they are both concrete structures, whereas LiDAR data can effectively distinguish them due to elevation and height information. In contrast, only LiDAR data face challenges in accurately distinguishing different materials of the same height (e.g., lawn and road). Therefore, the integration of hyperspectral and LiDAR data is widely used for land-cover and land-use classification [6–8].

In recent years, how to effectively integrate hyperspectral and LiDAR data has become a hot research topic. The current approaches can be categorized into three types: pixel-level fusion, feature-level fusion and decision-level fusion. Pixel-level fusion directly synthesizes the hyperspectral and LiDAR data at the input layer and then performs classification [9–11]. However, pixel-level fusion generally entails a significant amount of computational overhead, and it also exhibits a general low robustness due to some disturbances, such as noise. To tackle the aforementioned challenges and optimize the synergistic information between hyperspectral and LiDAR data, feature-level fusion techniques have been extensively employed. In [12], the classification is achieved by combining hyperspectral and LiDAR features extracted through Extended Attribute Profiles (EAPs). However, the mere concatenation or superposition of their features may result in redundant information and impede classification performance. The principal component analysis (PCA) is an effective method for reducing redundancy by mapping high-dimensional data onto an orthogonal low-dimensional space [13]. Meanwhile, Du et al. [14] proposed a graph fusion approach that leverages the correlation between hyperspectral and lidar data to integrate their features. Furthermore, a manifold alignment approach is proposed in [15] to enhance the acquisition of shared features across diverse modalities and integrate them with modality specific features for classification purposes. Besides feature-level fusion, decision-level fusion is also a widely adopted approach. In [16], it was proposed to use the support vector machine (SVM) for each feature, and then use Naive Bayes to fuse classifiers to obtain classification results. In [17], the maximum likelihood classifier (MLC), SVM and multinomial logistic regression classifiers were used to classify the features, and weighted voting was then used to obtain the final classification results. Although both feature-level and decision-level fusion techniques are capable of integrating information from different modalities, they often heavily rely on human experience in parameter selection and feature designment. Moreover, they also face the challenge of achieving a balance between the algorithmic precision and generalizability.

With the increasing popularity of deep learning in remote sensing [18–21], a plethora of fusion algorithms have been proposed. Among them, the convolutional neural network (CNN) has proven to be effective in extracting deep features from images and is widely utilized for the joint classification of hyperspectral and LiDAR data. In [22], a two-branch CNN is proposed to extract the spatial and spectral features of hyperspectral data, respectively, and then fuse with the features of LiDAR data for classification. The space and spectral characteristics of hyperspectral data are effectively utilized. Hang et al. [23] tried to use the CNN to extract the features of hyperspectral and LiDAR data in the way of parameter sharing, and after the feature-level fusion, the weighted summing method was adopted at the decision level to obtain the final classification results. To better utilize the complementary information between hyperspectral and LiDAR data, many researchers have proposed some methods. For instance, Zhang et al. [24] proposed a bidirectional autoencoder for hyperspectral and LiDAR data fusion, which utilized spectral and texture metrics to constrain the structure of the fused information, while integrating it using the Gram matrix. The resulting fusion outputs were then fed into a two-branch CNN for classification, reducing the reliance on training samples by leveraging complementary information. In [25], a multi-branch fusion network of self-and cross-guided attention is proposed. Specifically, the LiDAR-derived attention mask guides both the HS and the LiDAR itself. Meanwhile, it is sent to the fusion module together with the spectral supplement module for classification fusion. This approach can effectively interact with complementary information from different modalities. Similarly, Fang et al. [26] proposed a spatial-spectral enhancement module that effectively enhances the interaction between hyperspectral and LiDAR modalities by enhancing the spatial features of hyperspectral data with LiDAR features and enhancing the spectral information of LiDAR with hyperspectral features. In [27], Wang et al. proposed a three-branch CNN backbone network that can simultaneously extract spectral, spatial and elevation information. They utilized

hierarchical fusion to achieve a feature interaction and the integration of hyperspectral and LiDAR data, resulting in significant improvements in classification accuracy.

Although the above models have achieved acceptable results, most existing methods extract features of different modalities separately, and then directly integrate the complementary information. Intuitively, the complementary information highlights the distinctiveness of each modality, and the direct integration of this unique information leads to suboptimal fusion performance. Shared features across modalities can demonstrate their affinity and facilitate smoother connections between them, serving as a "bridge" to alleviate huge modality differences. Therefore, the integration of complementary and shared features across modalities is essential for their mutual enhancement. By interactively integrating these features, information from different modalities can be optimally combined. Inspired by this, our insight is the first key research question (RQ), **RQ1: How can we effectively utilize the shared-complementary information of hyperspectral and LiDAR images?** In addition, feature-level fusion enables the comprehensive processing of information, including edges, shapes, contours and local features, while decision-level fusion exhibits excellent error correction capabilities. Generally, the strategy of feature-level and decision-level fusion play distinct yet equally important roles in improving the performance of hyperspectral and LiDAR classification. However, the current feature-level and decision-level joint classification methods exhibit limited adaptability and flexibility. Therefore, the next key research question is raised: **RQ2: How can we adaptively collaboratively integrate the strategy of feature-level and decision-level fusion?**

In this paper, **to address the above RQ1**, we propose a multilevel interactive fusion (MIF) module to integrate the shared-complementary information of HS and LiDAR data while weakening the differences between the modalities. Specifically, the MIF module is cascaded by several three-branch feature interaction (TBFI) modules, which introduces an intermediate state branch in addition to the HS and LiDAR branches, with the aim of reducing modal differences and enabling the full interaction and integration of shared and complementary information through multiple TBFI modules. **To tackle the above RQ2**, the dynamic weight selection (DWS) module in decision-level fusion is attentively designed, which takes the three output layers of the MIF module as the input and adaptively assigns weights to the features obtained in feature-level fusion. The main contributions of this paper are summarized as follows:

(1) A feature-decision level collaborative fusion network for hyperspectral and LiDAR classification was proposed, which collaboratively integrates the strategy of feature-level and decision-level fusion to a unified framework.

(2) In feature-level fusion, an MIF module was developed that incorporates a novel intermediate state branch to optimize the utilization of shared-complementary features between hyperspectral and lidar data while minimizing mode discrepancies. This branch facilitates the information's interaction and integration with both the HS and Lidar branches within the TBFI module, utilizing a multi-level enhancement method to extract and transfer shared-complementary features between the two streams.

(3) In decision-level fusion, a DWS module was proposed to adaptively optimize the feature representations of the three branches by dynamically weighting their feature outputs to ensure a balanced and effective feature representation.

The rest of this article is organized as follows. The proposed framework is introduced in detail in Section 2. In Section 3, the experimental configuration and parameter setting, the classification results and analysis and ablation studies are given. Finally, a summary is given in Section 4.

## 2. Materials and Methods

### 2.1. Overall Framework of the Proposed Model

In this paper, a novel Feature-Decision level Collaborative Fusion Network (FDCFNet) was proposed for hyperspectral and LiDAR classification; the overall framework is shown in Figure 1. The input mainly consists of three branches: hyperspectral, intermediate

state and LiDAR. At the feature level, considering directly integrating the complementary HS and LiDAR features may lead to suboptimal fusion performance, due to the distinct modality difference. The shared features between HS and LiDAR can show the affinity between them, promote a smoother connection and become a "bridge" to alleviate the huge modality differences. Therefore, integrating the complementary and shared characteristics of the HS and LiDAR is essential for their mutual enhancement. Inspired by this, we designed a multilevel interaction fusion module and then three-branch feature interaction (TBFI) modules to effectively learn the complementary and shared features of hyperspectral and LiDAR data while reducing the difference between modalities. At the decision level, we designed the DWS module to optimize the information representation by dynamically assigning weights to three outputs of the MIF module. Most of the existing models often simply combine the feature representations together, or stack them together with some fully connected layers [28–30]. However, a mere stacking or fusion of these features may lead to an imbalanced representation of information, thereby diminishing the classification accuracy. Therefore, based on feature-level fusion, we further introduce a decision-level fusion component, i.e., the DWS module, which can dynamically allocate weights to the output from three branches in order to balance the representation of information from different modalities.

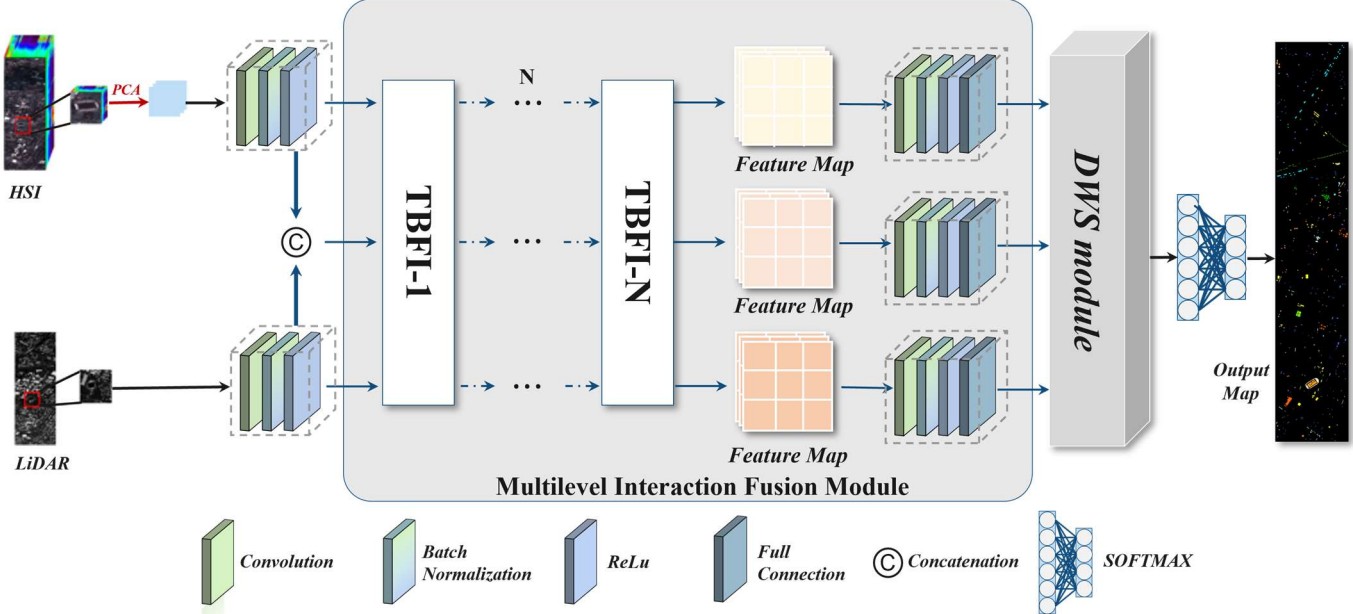

**Figure 1.** The overall framework of FDCFNet.

### 2.2. Multilevel Interaction Fusion Module

The multilevel interaction fusion module is shown in Figure 2, given a hyperspectral image $I_H$ and its corresponding LiDAR image $I_L$. First of all, in order to eliminate redundant information in the hyperspectral image, PCA was used to reduce the number of bands in hyperspectral images to $C$, where $C$ is the first $C$ principal components. Then, select a cube $P_H \in R^{M \times N \times C}$ in $I_H$, and select its corresponding patch $P_L \in R^{M \times N \times 1}$ in $I_L$, where M, N and $C$ represent the height, width and band number of the patch, respectively. The $P_H$ and $P_L$ are fed into a convolution layer to initially learn features, followed by the batch normalization (BN) layer and the rectified linear unit (Re*LU*) to regularize and learn nonlinearities. The output results are $F_H$ and $F_L$, respectively. In order to reduce the impact of the modality difference in the interaction of complementary information, we added an intermediate branch $F_f$, indirectly connecting the HS and LiDAR branches, which can better realize the joint learning of shared and complementary features. It can be formulated by

$$F_f = concat(F_H, F_L), \tag{1}$$

where $concat(\cdot)$ represents the combinatorial operation. After that, $F_H$, $F_f$ and $F_L$ are inputs to the TBFI module. In order to further mine and integrate the multi-level features while reducing modality differences, the N TBFI modules are sequentially employed, with the output of each preceding module serving as the input to the subsequent one, and by analogy, the $F_N^H$, $F_N^f$ and $F_N^L$ are assumed to be the outputs of the N-level TBFI modules, which are formulated as

$$\begin{cases} F_N^H = g_N^{TBFI}, \cdots, g_i^{TBFI}, \cdots, g_1^{TBFI}(F_H) \\ F_N^f = g_N^{TBFI}, \cdots, g_i^{TBFI}, \cdots, g_1^{TBFI}\left(F_f\right) \\ F_N^L = g_N^{TBFI}, \cdots, g_i^{TBFI}, \cdots, g_1^{TBFI}\left(F_f\right) \end{cases} \tag{2}$$

where $F_H$, $F_f$ and $F_L$ represent the three inputs, $g^{TBFI}$ represents the function of the TBFI module and $i$ represents the function of the $i$ th TBFI module layer.

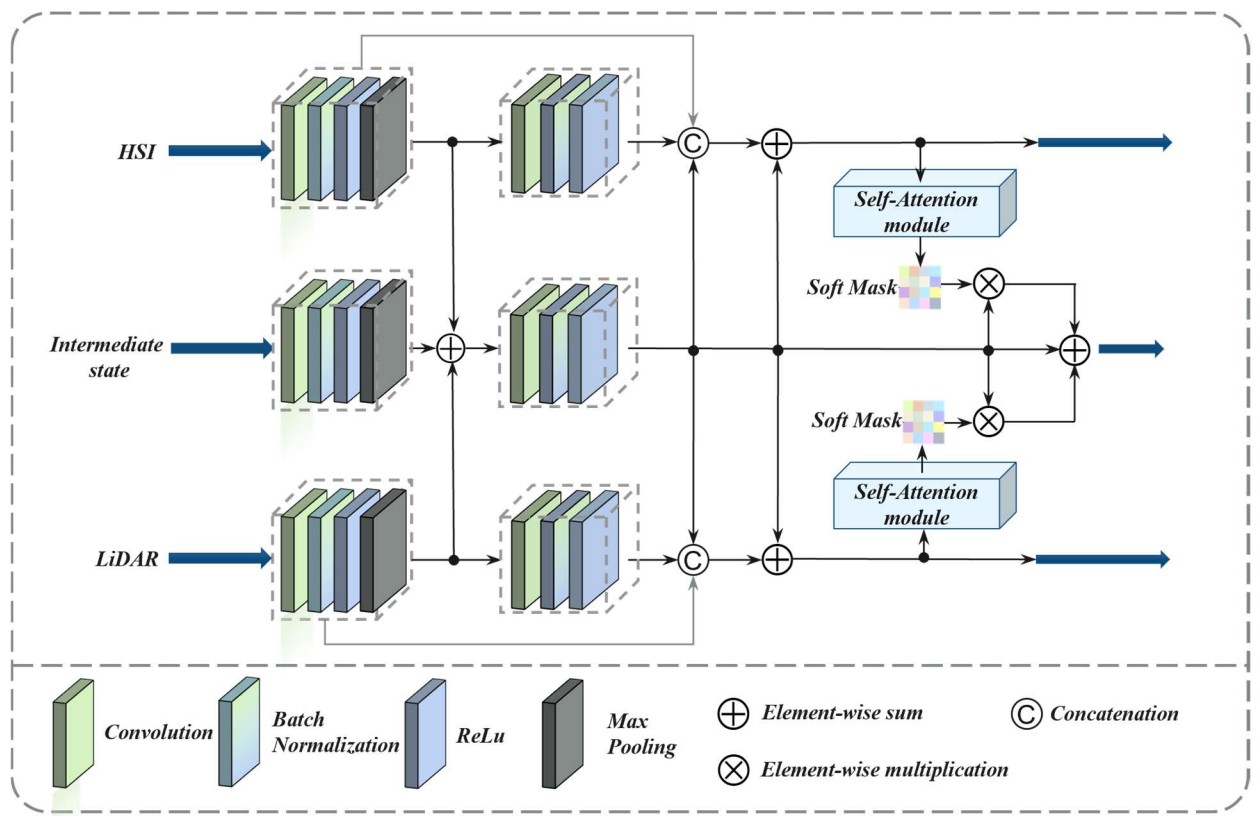

**Figure 2.** The structure of the three-branch feature interaction module.

The details of the TBFI module are shown in Figure 2. Take the first TBFI module layer as an example. First, the input feature maps of the three branches are fed to the convolutional layer to obtain the standardized feature maps $\overset{\wedge}{F}_H$, $\overset{\wedge}{F}_f$, $\overset{\wedge}{F}_L$:

$$\begin{cases} \overset{\wedge}{F}_H = Nconv(F_H) \\ \overset{\wedge}{F}_f = Nconv\left(F_f\right) \\ \overset{\wedge}{F}_L = Nconv(F_L) \end{cases} \tag{3}$$

where *Nconv* includes the convolution operation, BN layer, Re*LU* activation function and max-pooling layer. Among them, $\overset{\wedge}{F}_H$, $\overset{\wedge}{F}_f$ and $\overset{\wedge}{F}_L$ contain rich spectral information, spatial information and elevation information, respectively. Then, in order to further obtain their shared and complementary features, the spectral information extracted from the HS branch

and the elevation information extracted from the LiDAR branch are simultaneously injected into the intermediate state branch, and the resulting feature map $\hat{F}_f^s$ is obtained:

$$\hat{F}_f^s = \hat{F}_f \oplus \hat{F}_H \oplus \hat{F}_L \tag{4}$$

where $\oplus$ represents the addition operation of elements. Then, to further supplement the complementary information between the two modalities and enhance the shared information, the three branches carry out the feature interaction between modalities through the intermediate branches after the convolution operation, BN layer and *ReLU* activation function. The procedure can be formulated as

$$\begin{cases} F_1^H = \hat{F}_H \oplus concat\left[Bconv\left(\hat{F}_H\right), Bconv\left(\hat{F}_f^s\right)\right] \\ F_1^L = \hat{F}_L \oplus concat\left[Bconv\left(\hat{F}_L\right), Bconv\left(\hat{F}_f^s\right)\right] \end{cases} \tag{5}$$

and among them, $F_1^H$ and $F_1^L$ represent the feature maps of the HS and LiDAR branches, respectively, which are supplemented with complementary information and enhanced by shared information, that is, both branches fully contain spectral, spatial and elevation information. *Bconv* includes the convolution operation, BN layer and *ReLU* activation function. Finally, to further explore the shared high-level features, the feature maps of both the HS and LiDAR branches are fed into the self-attention module in order to generate attention masks. These masks are then injected into the middle branch to enhance focus on the common information; the specific formula is

$$F_1^f = Bconv\left(\hat{F}_f^s\right) \otimes \left[1 \oplus SA\left(F_1^H\right) \oplus SA\left(F_1^L\right)\right] \tag{6}$$

where $\otimes$ represents the multiplication operation of elements, and $SA(\cdot)$ represents the operation in which features pass through the self-attention module to generate an attention mask. The self-attention module is shown in Figure 3. Firstly, the matrix multiplication operation is performed on the input feature $W$ and its transpose $W^T$, and then the result is fed into the softmax layer for normalization to obtain the attention mask.

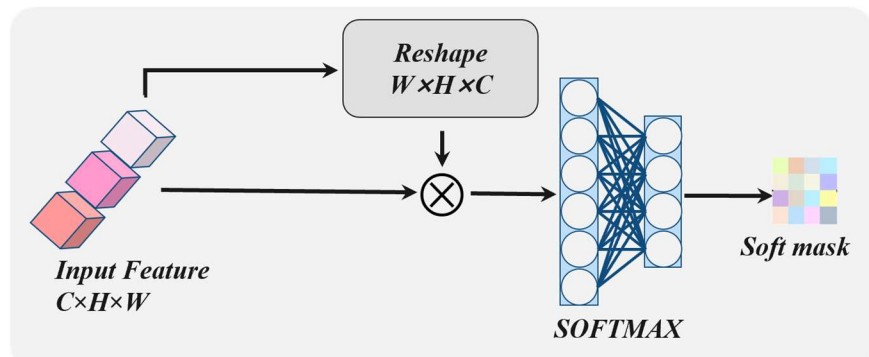

**Figure 3.** Illustration of self-attention module; $\otimes$ denotes element-wise multiplication.

Finally, the output feature maps $F_N^H$, $F_N^f$ and $F_N^L$ after the N-level TBFI modules are fed to the fully connected layer after the convolution operation, BN layer and ReLU activation function, and the outputs $D_H$, $D_f$, $D_L$ can be formulated as

$$\begin{cases} D_H = FC\left(F_N^H\right) \\ D_f = FC\left(F_N^f\right) \\ D_L = FC\left(F_N^L\right) \end{cases} \tag{7}$$

where *FC* represents the operation of feeding features to the fully connected layer for high integration.

### 2.3. Dynamic Weight-Selecting Module

The proposed dynamic weight-selecting (DWS) module is shown in Figure 4. In order to better integrate the information output by the MIF module, $D_H$, $D_f$ and $D_L$ are first combined to obtain the sum of the information represented, which can be formulated as

$$S = D_H \oplus D_f \oplus D_L. \tag{8}$$

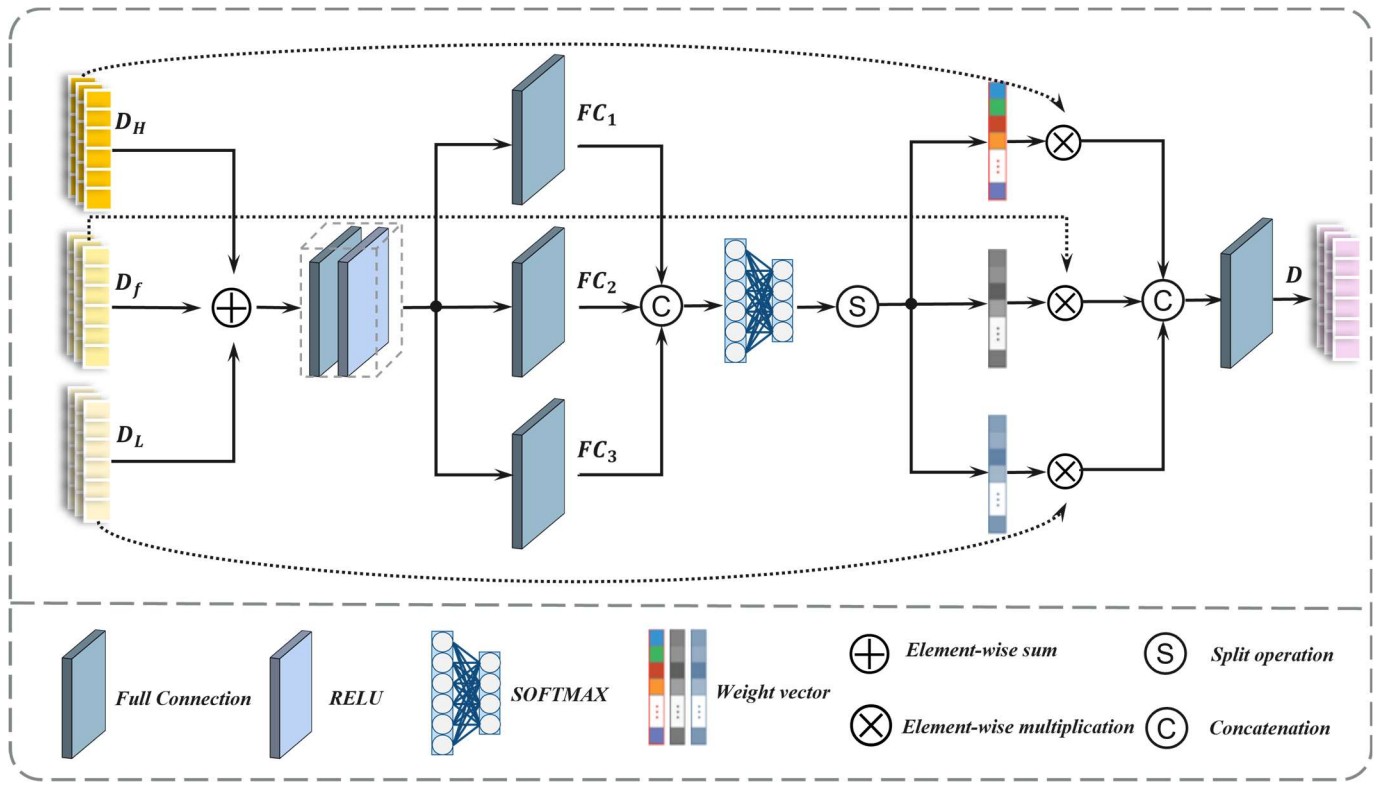

**Figure 4.** Illustration of self-attention.

Then, $S$ is fed to the full connection layer and Re*LU* activation function successively to learn more about the representation of information. After that, the output is placed separately into three fully connected layers, aiming to learn the representation of the information with different weights. Then, the output is integrated through the concat function, which can be specifically expressed as

$$S_1 = \mathrm{Re}LU(FC(S)) \tag{9}$$

$$S_2 = concat(FC_1(S_1) + FC_2(S_1) + FC_3(S_1)) \tag{10}$$

where *FC* represents the first fully connected layer; Re*LU*($\cdot$) represents learning nonlinearity through the Re*LU* activation function; $FC_1$, $FC_2$ and $FC_3$ represent three fully connected layers with different weights; $S \in R^{Q \times 1}$, $S_1 \in R^{Q \times 1}$ and $S_2 \in R^{3Q \times 1}$; and $Q$ represents the number of classes of land-covers to be distinguished. Then, the weight selection operation is carried out on the output information representation $S_2$:

$$w_H, w_f, w_L = SP(softmax(S_2)) \tag{11}$$

and among them, $softmax(\cdot)$ represents the softmax function and $SP(\cdot)$ represents the split operation, which evenly divides the input into three parts. $w_H$, $w_f$ and $w_L$ are the weight distribution vectors. Finally, the output of the DWS module can be expressed as

$$D = w_H \otimes D_H + w_f \otimes D_f + w_L \otimes D_L. \tag{12}$$

*2.4. Classification and Loss Function Design*

The output $D$ of the DWS module is the preliminary classification map, and the classification result $O$ can be obtained by the following calculation:

$$O = \mathrm{argmax}(softmax(D)) \tag{13}$$

where $\mathrm{argmax}(\cdot)$ represents the argmax function, which gives the class with the maximum probability.

We send the training set into the network in the form of $\left\{ X_H^i, X_L^i, Y^i \middle| i = 1, 2, \cdots, N \right\}$, where $\left\{ X_H^i, X_L^i \right\}$ and $Y^i$ represent the input and corresponding class label value, respectively, $i$ represents the $i$ th sample and $K$ represents the aggregate number of samples. Moreover, we employ the cross-entropy loss function to compute the loss value of the predicted output. $\overset{\wedge}{Y}$ and the class label value $Y$ show where the predicted value represents the probability that the output result is the class label value. The specific calculation formula is shown as

$$loss = -\frac{1}{K}\sum_{i=1}^{K} Y^i \log\left( \overset{\wedge}{Y^i} \right). \tag{14}$$

Simultaneously, for enhanced supervision of the feature learning process across the three branches of FDCFNet, the loss value was calculated for the output of the three branches to obtain $loss_H$, $loss_f$ and $loss_L$, which, respectively, represent the loss function of the hyperspectral, intermediate state and LiDAR branch. And the loss function $l$ of the whole frame is designed as

$$l = \mu_1 loss_H + \mu_2 loss_f + \mu_3 loss_L + loss \tag{15}$$

where among them, $\mu_1$, $\mu_2$ and $\mu_3$, respectively, represent the weight parameters of $loss_H$, $loss_f$ and $loss_L$. Based on the experience of a large number of experiments, we set them as 0.001, 0.01 and 0.001, respectively.

**3. Experiment and Result**

Three benchmark remote sensing (RS) datasets were utilized as opening references to assess the performance of FDCFNet. First, the experimental datasets were described. Secondly, the details of the experiments were given. After that, we compared and analyzed the classification results of FDCFNet with different comparison methods. Finally, we conducted ablation experiments to demonstrate the efficacy of various modules.

*3.1. Experimental Datasets*

In this paper, three common RS datasets: the Houston dataset [31,32], Trento dataset [33] and MUUFL dataset [34,35], are selected to evaluate the performance of FDCFNet. The land-cover object classes and corresponding numbers of training and testing samples for the three datasets are presented in Tables 1–3. A detailed description of these datasets follows.

(1) Houston dataset: The dataset comprises a hyperspectral image with 144 spectral bands and a wavelength range of 0.38 to 1.05 μm, as well as LiDAR data, capturing the University of Houston campus at a spatial resolution of 2.5 m over an area size of $349 \times 1905$ pixels. The LiDAR data contains one band, while the training and testing labels encompass fifteen classes.

**Table 1.** Number of training and testing samples for the Houston dataset.

| No. | Class | Number of Train Samples | Number of Test Samples |
|:---:|:---:|:---:|:---:|
| 1 | Health grass | 198 | 1251 |
| 2 | Stressed grass | 190 | 1254 |
| 3 | Synthetic grass | 192 | 697 |
| 4 | Trees | 188 | 1244 |
| 5 | Soil | 186 | 1242 |
| 6 | Water | 182 | 325 |
| 7 | Residential | 196 | 1268 |
| 8 | Commercial | 191 | 1244 |
| 9 | Road | 193 | 1252 |
| 10 | Highway | 191 | 1227 |
| 11 | Railway | 181 | 1235 |
| 12 | Parking lot 1 | 192 | 1233 |
| 13 | Parking lot 2 | 184 | 469 |
| 14 | Tennis court | 181 | 428 |
| 15 | Running track | 187 | 660 |
| | Total | 2832 | 15,029 |

**Table 2.** Number of training and testing samples for the Trento dataset.

| No. | Class | Number of Train Samples | Number of Test Samples |
|:---:|:---:|:---:|:---:|
| 1 | Apple trees | 129 | 4034 |
| 2 | Building | 125 | 2903 |
| 3 | Ground | 105 | 479 |
| 4 | Woods | 154 | 9123 |
| 5 | Vineyard | 184 | 10,501 |
| 6 | Roads | 122 | 3174 |
| | Total | 819 | 30,214 |

**Table 3.** Number of training and testing samples for the MUUFL dataset.

| No. | Class | Number of Train Samples | Number of Test Samples |
|:---:|:---:|:---:|:---:|
| 1 | Trees | 150 | 23,246 |
| 2 | Mostly grass | 150 | 4270 |
| 3 | Mixed ground surface | 150 | 6882 |
| 4 | Dirt and sand | 150 | 1826 |
| 5 | Road | 150 | 6687 |
| 6 | Water | 150 | 466 |
| 7 | Building Shadow | 150 | 2233 |
| 8 | Building | 150 | 6240 |
| 9 | Sidewalk | 150 | 1385 |
| 10 | Yellow curb | 150 | 183 |
| 11 | Cloth panels | 150 | 269 |
| | Total | 1650 | 53,687 |

(2) Trento dataset: The second dataset covers a rural region situated to the south of Trento, Italy. The hyperspectral image was acquired by the AISA Eagle sensor, which features 63 spectral bands spanning from 0.42 to 0.99 µm in wavelength, while the LiDAR data was collected using the Optech ALTM 3100EA sensor and consists of one spectral band. This dataset measures at a spatial resolution of 1 m with dimensions measuring at 166 × 600 pixels. The training and testing labels contain six distinct categories.

(3) MUUFL dataset: The third dataset has a size of 325 × 220 pixels and was taken at the University of Mississippi Gulf Coast campus. It contains a stream of hyperspectral imagery and LiDAR data, where the hyperspectral image has 64 spectral bands with

wavelengths ranging from 0.38 to 1.05 μm; LiDAR data have two spectral bands. The training and testing labels contain 11 different categories.

### 3.2. Experimental Configuration

To assess the performance of different methods, all experiments were conducted on a PC equipped with an Intel(R) Xeon(R) Gold 5218 CPU, operating at 2.30 GHz, an NVIDIA Quadro P5000 GPU, 32 GB of RAM and Windows 10. Our proposed programs were written in the PyCharm compiler with python3.8, and some deep learning networks were implemented using the PyTorch framework. At the same time, we use the Adam optimization algorithm [36] to adaptively optimize these models. Set the batch size and epoch to 64 and 150, respectively, for training.

In this paper, three metrics were employed—average accuracy (AA), overall accuracy (*OA*) and the kappa coefficient (kappa)—to assess the performance. The *OA* represents the proportion of correctly classified pixels to total classified pixels. The AA denotes the mean classification accuracy across all categories, while the kappa measures error reduction between classification and random guessing. *Kappa* can be mathematically expressed as follows:

$$Kappa = \frac{OA - p_e}{1 - p_e} \times 100\%$$ (16)

where

$$p_e = \frac{\sum\limits_{i=1}^{Q} a_i \times b_i}{n \times n},$$ (17)

where $Q$ is the number of classes, $a_i$ is the number of actual samples of the $i$ th class, $b_i$ is the number of predicted samples of the $i$ th class and n represents the aggregate quantity of samples to be classified. These performance indicators can assess the classification accuracy of the model and a higher value indicates a superior classification performance.

### 3.3. Classification Results and Analysis

In order to verify the validity of FDCFNet, we compare the proposed model with several state-of-the-art methods, including the support vector machine (SVM) [37], random forest (RF) [38], shared and specific feature learning model (S2FL) [14], common subspace learning (CoSpace-ℓ1) [39], Two-branch convolution neural network (two-branch CNN) [21], Coupled CNN [22], Multi-attentive hierarchical dense fusion net (MAHiDFNet) [26] and Spatial–spectral cross-modal enhancement network(S2ENet) [25]. Meanwhile, in order to ensure comparability across experiments, identical training and testing samples were used.

Tables 4–6 show the classification results on the three datasets, i.e., Houston, Trento and MUUFL, respectively. The visual classification maps of the three datasets are presented in Figures 5–7 for comparison among different methods.

**Table 4.** Classification accuracy of the Houston data by different methods.

| No. | Class | SVM | RF | S2FL | CoSpace-ℓ1 | Two-Branch CNN | Coupled CNN | MAHiDFNet | S2ENet | Proposed |
|-----|-------|-----|-----|------|------------|----------------|-------------|-----------|--------|----------|
| 1 | Health grass | 94.06 | 92.75 | 90.57 | 89.45 | 98.80 | 85.61 | 98.53 | 86.81 | 97.45 |
| 2 | Stressed grass | 95.07 | 95.39 | 97.69 | 97.05 | 84.22 | 100.00 | 92.87 | 100.00 | 100.00 |
| 3 | Synthetic grass | 100 | 95.12 | 100.00 | 100.00 | 96.27 | 97.85 | 91.11 | 100.00 | 96.84 |
| 4 | Trees | 98.51 | 99.01 | 98.39 | 98.47 | 95.22 | 99.92 | 98.10 | 99.92 | 99.60 |
| 5 | Soil | 95.77 | 96.02 | 99.36 | 99.36 | 97.56 | 100.00 | 98.38 | 100.00 | 100.00 |
| 6 | Water | 79.37 | 82.54 | 99.38 | 99.69 | 98.33 | 100.00 | 95.58 | 99.69 | 100.00 |
| 7 | Residential | 92.44 | 90.41 | 80.52 | 80.52 | 98.01 | 91.01 | 99.15 | 93.68 | 94.64 |
| 8 | Commercial | 80.53 | 68.35 | 68.57 | 89.39 | 86.34 | 93.73 | 80.94 | 95.82 | 97.11 |
| 9 | Road | 79.9 | 70.02 | 69.89 | 64.86 | 74.04 | 94.73 | 98.04 | 88.98 | 94.68 |
| 10 | Highway | 93.78 | 78.49 | 70.01 | 66.75 | 90.05 | 91.93 | 72.81 | 92.67 | 94.38 |
| 11 | Railway | 88.31 | 82.05 | 88.34 | 88.42 | 92.44 | 96.60 | 72.71 | 98.06 | 98.30 |

**Table 4.** *Cont.*

| No. | Class | SVM | RF | S2FL | CoSpace-ℓ1 | Two-Branch CNN | Coupled CNN | MAHiDFNet | S2ENet | Proposed |
|---|---|---|---|---|---|---|---|---|---|---|
| 12 | Parking lot 1 | 62.12 | 67.56 | 84.27 | 84.02 | 57.50 | 94.24 | 76.80 | 87.10 | 93.03 |
| 13 | Parking lot 2 | 37.14 | 12.75 | 83.58 | 78.68 | 100.00 | 95.74 | 95.80 | 94.03 | 91.82 |
| 14 | Tennis court | 98.31 | 87.47 | 100.00 | 100.00 | 96.39 | 99.07 | 99.53 | 100.00 | 99.77 |
| 15 | Running track | 98.75 | 86.41 | 98.94 | 98.18 | 98.79 | 98.33 | 100.00 | 100.00 | 90.37 |
| | OA (%) | 87.61 | 82.50 | 86.81 | 87.52 | 87.18 | 95.34 | 89.58 | 95.09 | **96.61** |
| | AA (%) | 86.27 | 80.29 | 88.63 | 88.99 | 90.93 | 95.92 | 91.36 | 95.78 | **96.53** |
| | Kappa × 100 | 86.60 | 81.05 | 85.75 | 86.51 | 86.12 | 94.96 | 88.74 | 94.69 | **96.34** |

**Table 5.** Classification accuracy of the Trento data by different methods.

| No. | Class | SVM | RF | S2FL | CoSpace-ℓ1 | Two-Branch CNN | Coupled CNN | MAHiDFNet | S2ENet | Proposed |
|---|---|---|---|---|---|---|---|---|---|---|
| 1 | Apple trees | 81.91 | 78.69 | 71.02 | 86.04 | 77.94 | 98.88 | 99.19 | 99.83 | 100.00 |
| 2 | Building | 91.82 | 85.01 | 82.57 | 95.83 | 89.29 | 97.83 | 88.92 | 98.21 | 98.29 |
| 3 | Ground | 93.55 | 94.62 | 92.07 | 95.62 | 70.03 | 97.49 | 97.53 | 100.00 | 99.48 |
| 4 | Woods | 98.07 | 94.86 | 86.69 | 98.73 | 100.00 | 99.89 | 99.98 | 99.96 | 100.00 |
| 5 | Vineyard | 92.11 | 90.21 | 48.90 | 63.56 | 99.60 | 100.00 | 99.90 | 99.84 | 100.00 |
| 6 | Roads | 80.55 | 77.75 | 77.57 | 89.16 | 97.74 | 92.82 | 99.78 | 90.93 | 93.23 |
| | OA (%) | 90.36 | 88.34 | 70.20 | 83.48 | 94.18 | 98.82 | 98.59 | 98.78 | **99.11** |
| | AA (%) | 88.00 | 86.86 | 76.47 | 88.16 | 89.10 | 97.82 | 97.55 | 98.13 | **98.50** |
| | Kappa × 100 | 87.13 | 84.40 | 61.91 | 78.66 | 92.32 | 98.42 | 98.12 | 98.38 | **98.81** |

**Table 6.** Classification accuracy of MUUFL data by different methods and the kappa coefficient.

| No. | Class | SVM | RF | S2FL | CoSpace-ℓ1 | Two-Branch CNN | Coupled CNN | MAHiDFNet | S2ENet | Proposed |
|---|---|---|---|---|---|---|---|---|---|---|
| 1 | Trees | 95.15 | 92.83 | 72.44 | 78.55 | 98.34 | 91.67 | 98.29 | 87.61 | 96.46 |
| 2 | Mostly grass | 74.34 | 78.47 | 68.67 | 74.36 | 81.54 | 94.70 | 86.63 | 85.76 | 84.29 |
| 3 | Mixed ground surface | 80.84 | 74.57 | 53.66 | 63.18 | 73.04 | 67.63 | 83.66 | 80.83 | 84.82 |
| 4 | Dirt and sand | 85.15 | 77.36 | 71.08 | 81.00 | 85.05 | 87.90 | 83.65 | 80.28 | 95.51 |
| 5 | Road | 93.05 | 89.01 | 58.47 | 77.99 | 84.54 | 88.26 | 96.67 | 83.67 | 89.43 |
| 6 | Water | 81.19 | 73.89 | 94.42 | 96.57 | 85.98 | 99.79 | 78.84 | 89.48 | 98.93 |
| 7 | Building Shadow | 65.6 | 69.99 | 76.62 | 83.65 | 80.29 | 97.49 | 74.85 | 95.25 | 96.15 |
| 8 | Building | 84.27 | 82.17 | 80.77 | 86.17 | 98.50 | 96.44 | 98.72 | 89.81 | 96.55 |
| 9 | Sidewalk | 48.25 | 35.89 | 51.05 | 63.25 | 78.28 | 91.77 | 71.93 | 88.66 | 87.65 |
| 10 | Yellow curb | 66.29 | 5.62 | 95.63 | 96.72 | 30.72 | 96.72 | 40.40 | 95.08 | 97.27 |
| 11 | Cloth panels | 77.39 | 86.59 | 97.40 | 98.88 | 81.13 | 99.26 | 64.43 | 99.63 | 99.63 |
| | OA (%) | 87.05 | 83.95 | 68.93 | 77.28 | 88.67 | 89.20 | 91.83 | 86.56 | **92.90** |
| | AA (%) | 77.41 | 69.4 | 74.56 | 81.85 | 79.76 | 91.97 | 79.82 | 88.73 | **93.33** |
| | Kappa × 100 | 82.81 | 78.82 | 61.35 | 71.30 | 85.19 | 86.03 | 89.31 | 82.81 | **90.68** |

Results on the Houston dataset: Our proposed method surpasses other recent influential deep learning-based methods, as demonstrated in Table 4, including the two-branch CNN, Coupled CNN, MAHiDFNet and S2ENet on the Houston dataset. The OA was increased by 9.43%, 1.27%, 7.03% and 0.83% respectively, while the kappa was increased by 10.22%, 1.38%, 7.6% and 1.65%. Among them, the accuracy of Commercial is much higher than that of the other classification methods, which is attributed to the multilevel interaction fusion module that can better interactively integrate the spectral information of hyperspectral data and the elevation information of LiDAR data. Specifically, it combines the unique information of the two modalities and enhances the common information of the two modalities. In Figure 5, it is evident that the proposed method yields a smoother visual effect and exhibits no classification errors or misjudgments when categorizing stressed grass, soil and water.

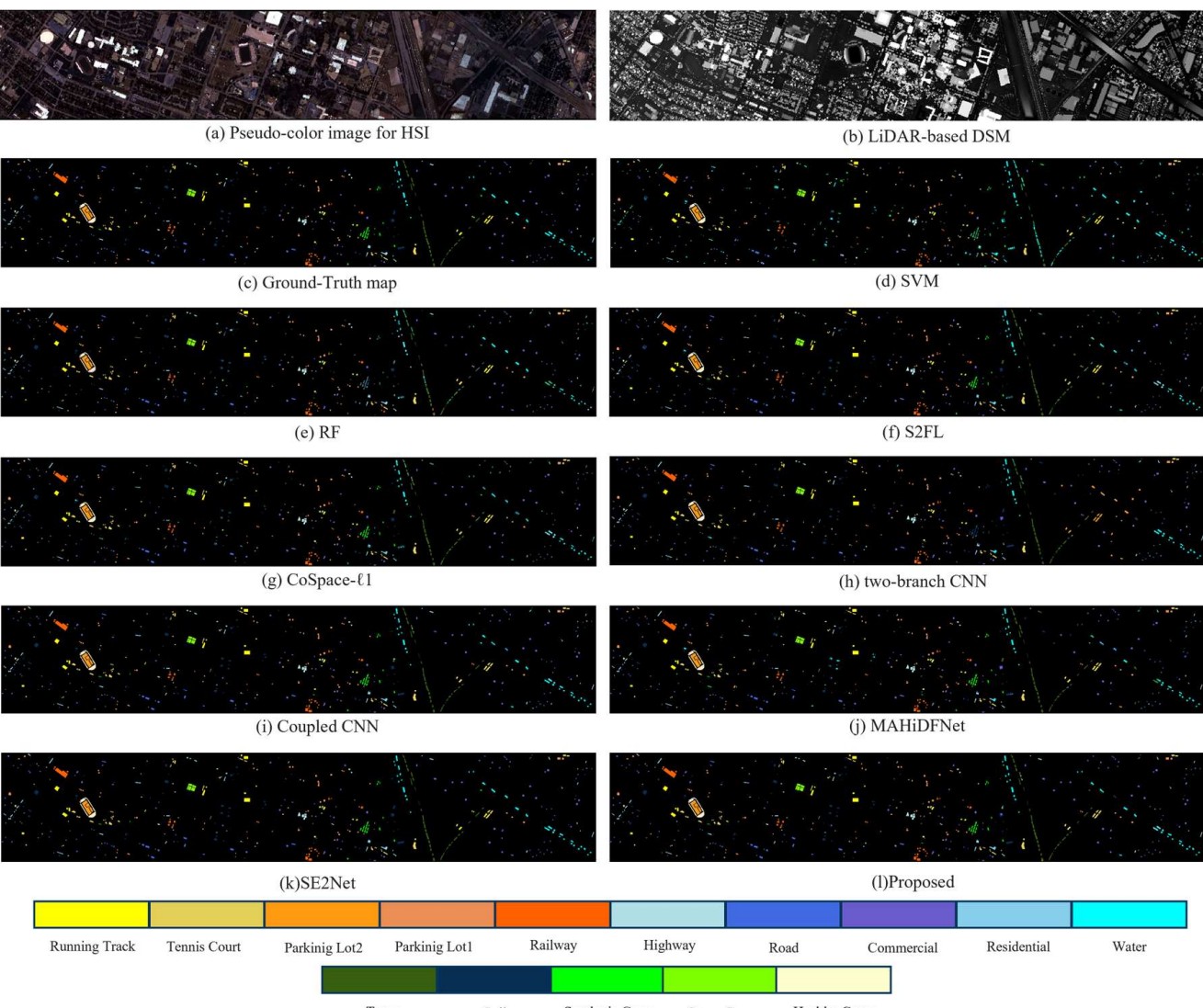

**Figure 5.** Classification maps of different methods on the Houston dataset. The pseudo-color image of HS image adopts the spectral bands of 45th, 25th and 10th as R, G and B.

Results on the Trento dataset: Table 5 shows the classification accuracy of the FDCFNet and other comparison methods on the Trento dataset; the OA of the proposed method achieves an outstanding 99.11%, in which the classification accuracy of apple trees, woods and vineyard reaches 100%. In Figure 6, it is evident that SVM and RF exhibit more misclassification patterns than certain deep learning algorithms. It may be because the single-input model will carry out multi-mode information fusion before input, which will cause certain information loss. Additionally, the classification results of the traditional methods S2FL and CoSpace-$\ell$1 are also unsatisfactory, possibly due to geometric information loss during the transformation of hyperspectral and LiDAR data into vectors. The FDCFNet exhibits a significant classification effect on large-scale objects, which can be attributed to the effective integration and interaction of complementary and shared information across different modalities facilitated by the MIF module. However, in the road classification, the effect is not satisfactory, which may be due to the phenomenon of misjudgment caused by information overlap when the common features and complementary features are fused and enhanced.

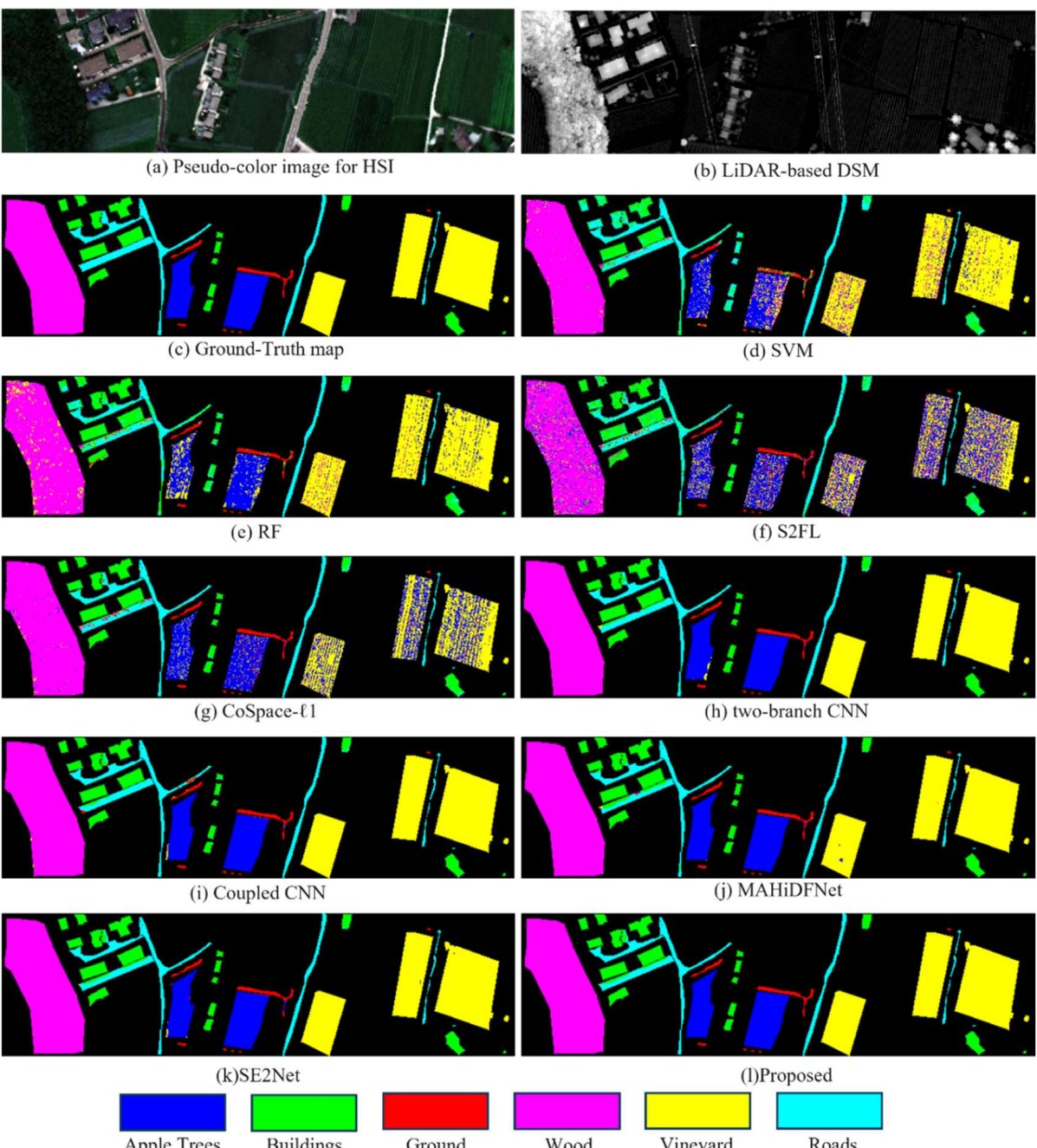

**Figure 6.** Classification maps of different methods on the Trento dataset. The pseudo-color image of the HS image adopts the spectral bands of 25th, 18th and 10th as R, G and B.

Results on the MUUFL dataset: Table 6 presents the classification accuracy of various methods on the MUUFL dataset. The proposed method exhibits a significantly superior classification performance compared to some classical comparison methods. The OA of FDCFNet surpasses that of several recently proposed deep learning methods, including the two-branch CNN, Coupled CNN, MAHiDFNet and S2ENet by 4.23%, 3.7%, 1.07% and 6.34%, respectively. In addition, we can also see the superiority of the proposed method in Figure 7. It can be seen that FDCFNet classifies almost all of the mixed ground surface and trees in the lower right corner of the figure correctly, and at the same time, the edges of trees and other ground objects are clear and very close to the ground truth which is attributed to the dynamic weight selection strategy, which can better balance the feature representation

of hyperspectral and LiDAR data, especially in the edge processing of classification. This fully demonstrates the superiority of the proposed model in land-cover classification.

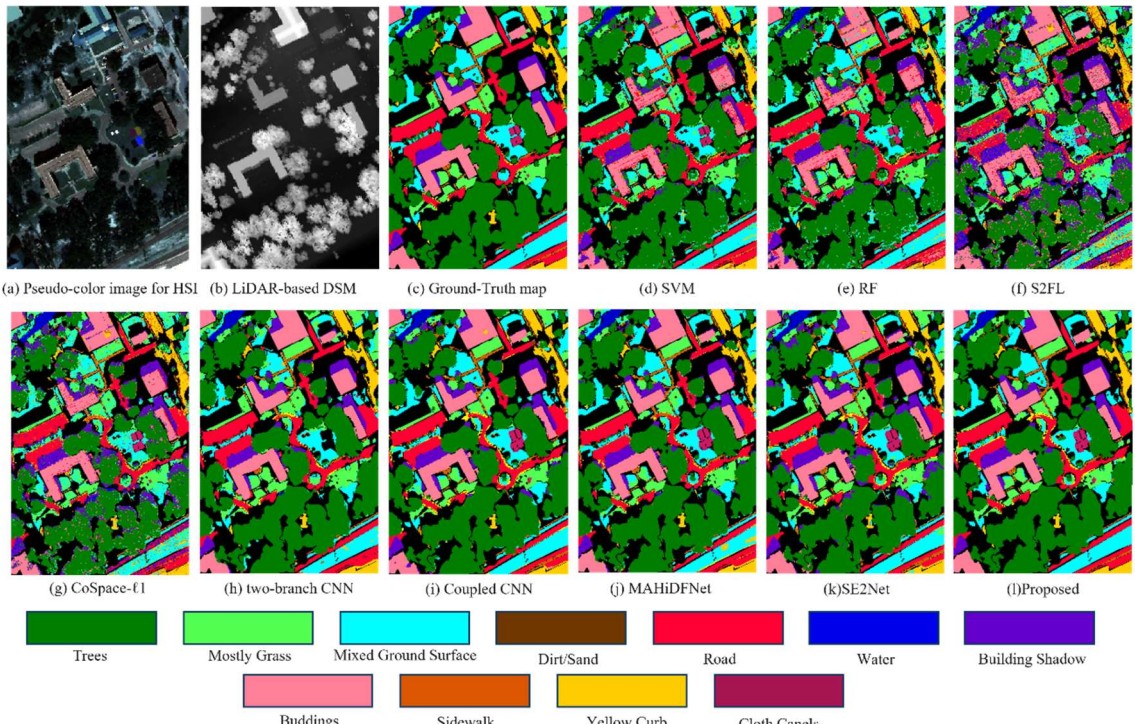

**Figure 7.** Classification maps of different methods on the MUUFL dataset. The pseudo-color image of HS image uses the 30th, 20th and 10th spectral bands as R, G and B.

On the whole, our proposed method outperforms other state-of-the-art methods, demonstrating a superior classification performance. The exceptional results of FDCFNet can be attributed to several key factors: firstly, an intermediate branch is introduced into the model to mitigate modality discrepancies. Secondly, multiple layers of the TBFI modules are employed for deep mining and the integration of shared-complementary features from hyperspectral and LiDAR data. Finally, a DWS module is incorporated to dynamically allocate weights to the output of three branches, ensuring a more equitable representation of information across different modalities. Among the three classification result graphs, FDCFNet exhibits a superior classification performance, with clearer boundaries and closer proximity to the ground truth. This further validates the advantages of our proposed model. In summary, our model demonstrates a high competitiveness in joint hyperspectral and LiDAR data classification.

## 4. Discussion

### 4.1. Parameter Tuning

As is widely acknowledged, the performance of the overall framework hinges on its submodules. Therefore, optimizing the overall performance by adjusting hyper-parameters in each submodule becomes a crucial issue to be addressed. Specifically, we fine-tune hyper-parameters such as the PCA dimensionality reduction, input patch size, network learning rate, number of TBFI modules in the MIF module and weight of loss function to maximize the overall performance. We assessed the classification performance of our models by computing OA, AA and kappa scores across various hyper-parameters. To ensure validity and reliability, we averaged the results over 10 experiments.

(1) Number of dimensionality reduction: Since the original hyperspectral data have a large number of spectral bands, it is easy to lead to the problem of dimensional disaster. The PCA can effectively alleviate this problem and minimize the loss of effective information

while reducing redundant information. The choice of dimensionality reduction $C$ has a certain influence on model performance. Therefore, other parameters were fixed and the sets of different $C\{1, 5, 10, 15, 20, 25, 30\}$ were tested on three RS data sets, respectively. Figure 8 shows the OA of different $C$ on the three data sets. It can be seen that in the three data sets, when $C$ is 25, the performance is the best. Therefore, we can set the dimensionality reduction $C$ to 25.

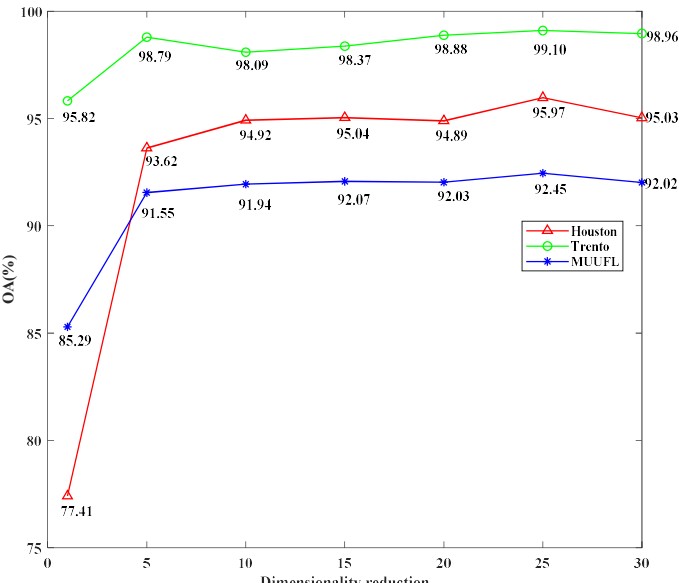

**Figure 8.** Effect of PCA dimension reduction on model performance (OA) by FDCFNet.

(2) Input patch size: The patch size of hyperspectral and LiDAR data determines the amount of information fed into the model. In order to avoid significant information loss during convolution, pooling and other operations, it is crucial to ensure that the input size is not too small. Conversely, if the input size is too large, there will be an increase in computational complexity without a corresponding increase in abstraction level. In order to determine the optimal network input patch size, while keeping other parameters fixed, patches ranging from $7 \times 7$ to $19 \times 19$ were discussed on three different remote sensing datasets. Figure 9 illustrates the overall accuracy (OA) of various patch sizes across these datasets, revealing that the model performs best with a patch size of $13 \times 13$.

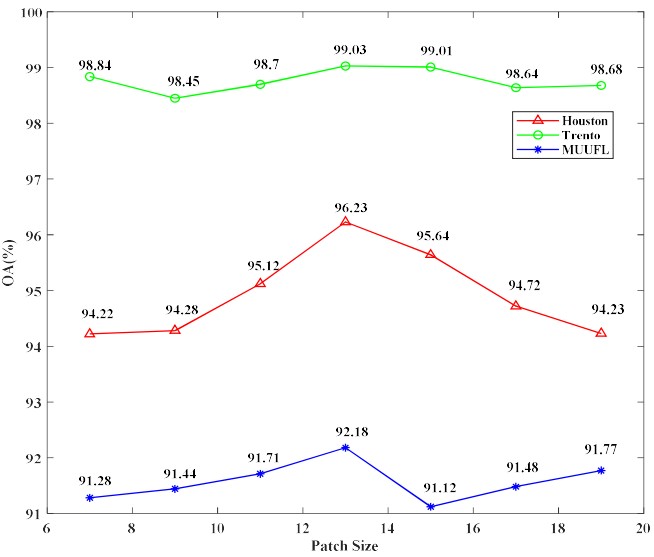

**Figure 9.** Effect of input patch size on model performance (OA) by FDCFNet.

(3) Network learning rate: The learning rate is a crucial hyper-parameter for updating parameters in neural networks, as it determines the convergence of the objective function to a local minimum and the time required for such a convergence. If the learning rate is too small, the network convergence becomes more complex and results in slow changes to the loss function. Conversely, if the learning rate is too high, there may be the direct skipping of local optima by the loss function leading to non-convergence. Therefore, a learning rate ranging from 0.0001 to 0.01 was analyzed for training on the three RS datasets in order to obtain the optimal value. As shown in Figure 10, a learning rate of 0.001 yielded the best results.

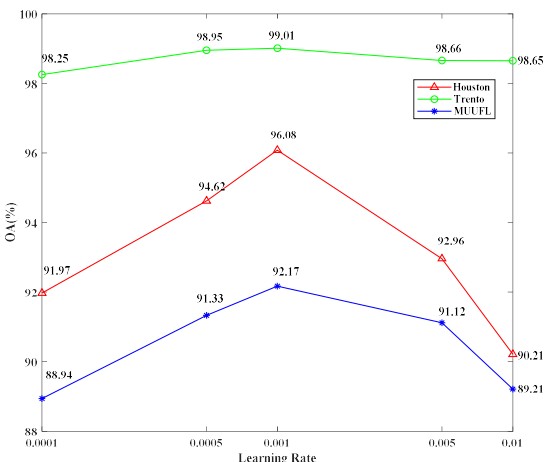

**Figure 10.** Effect of learning rate on model performance (OA) by FDCFNet.

(4) The number of TBFI modules: As discussed in Section 2.2, after simply extracting the features of the hyperspectral, intermediate state and LiDAR branch, in order to further excavate and integrate multi-level features and weaken the differences between modes, the N-level TBFI module is used to enhance their shared and complementary features. According to Equation (2), FDCFNet is affected by N. In order to discuss the influence of the number N of TBFI modules on model performance, we change the value of N on three data sets, respectively, to compare the classification results. Figure 11 shows the OA index of classification results of three data sets with different values of N. It is evident that the advanced shared and complementary features of the three branches cannot be effectively extracted when N equals 1, leading to a poor model performance. Conversely, if N is too large, overfitting may occur due to a limited number of training samples. The optimal value for N appears to be 2 as it yields the best model performance.

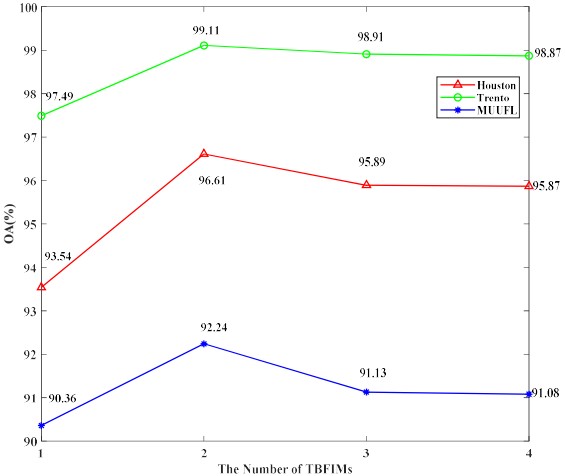

**Figure 11.** The effect of the number of layers of TBFI modules on the model performance.

(5) Weight parameter of the loss function: The loss function of the model is shown in Equation (15), where there are three hyperparameters: $\mu_1$, $\mu_2$ and $\mu_3$. In order to find the influence of these three hyperparameters on the model performance, we set different values in three data sets for experiments. Specifically, we first fix $\mu_2$ and $\mu_3$, change the value of $\mu_1$ from the $\{0.001, 0.01, 0.1, 1\}$ and set to find the $\mu_1$ that can achieve the best performance of the model. Then, we select the best $\mu_1$ and fix the value of $\mu_3$ and change the value of $\mu_2$ from the $\{0.001, 0.01, 0.1, 1\}$ set, so as to find the best $\mu_2$ value. Similarly, the optimal values of $\mu_1$ and $\mu_2$ are selected, and the values of $\mu_3$ are changed to find the value of $\mu_3$ that optimizes the model performance. Figure 12 shows the parameter experiments carried out on three data sets. The red line represents $\mu_1$, the yellow line represents $\mu_2$ and the blue line represents $\mu_3$. It can be seen from the three figures that when $\mu_1$, $\mu_2$ and $\mu_3$ are 0.001, 0.01 and 0.001, respectively, the model performance reaches the best.

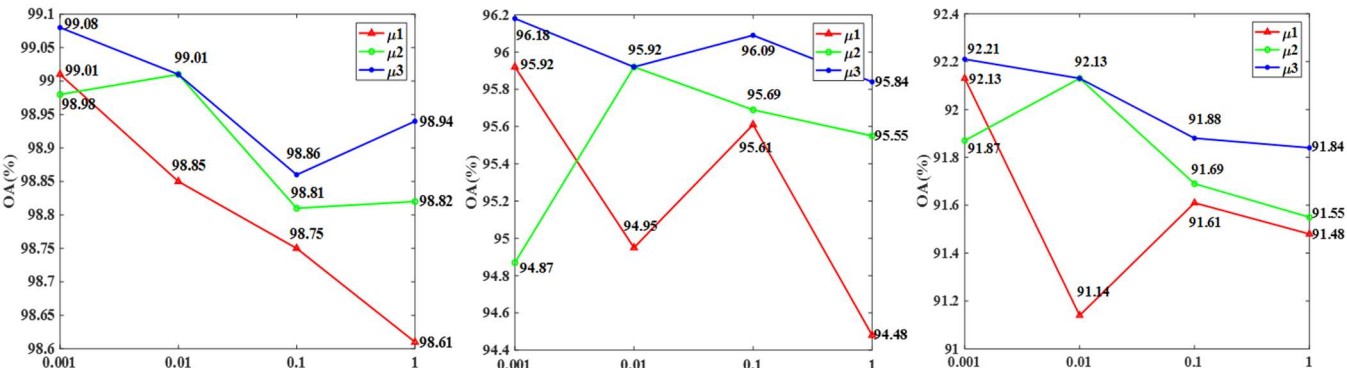

**Figure 12.** The influence of weight parameters $\mu_1$, $\mu_2$ and $\mu_3$ on the classification performance (OA) by FDCFNet on the three data sets: (Left to right) Houston data, Trento data and MUUFL data.

*4.2. Ablation Study*

In order to validate the efficacy of each model within the proposed framework for classification, we conducted ablation experiments on the intermediate state branch, TBFI module and DWS module, respectively, in this section. To ensure experimental reliability and fairness, all parameters were set to their optimal values, with OA, AA and kappa used as metrics for evaluating classification performance.

(1) Ablation study for intermediate state branch: In addition to the hyperspectral and LiDAR branches, we propose a novel intermediate state branch that indirectly connects the two modalities, reduces the mode gap, weakens modality differences and facilitates better interactive fusion of information from both hyperspectral and LiDAR branches. Table 7 demonstrates the classification performance of the model on three remote sensing datasets, both before and after incorporating an intermediate state branch. The results indicate a significant improvement in classification accuracy following the addition of this branch. In the Houston data set, OA has increased by 2.59%, and in the Trento data set, OA has increased by 3.08%. On the MUUFL dataset, OA improved by 2.19%. At the same time, AA and kappa also improved correspondingly, which indicates that the intermediate state branch can effectively improve the classification performance of the model.

**Table 7.** OA, AA and kappa coefficients of the model before and after adding the intermediate state branch in the three RS data sets.

| Dataset | Houston | | | Trento | | | MUUFL | | |
|---|---|---|---|---|---|---|---|---|---|
| | OA | AA | Kappa | OA | AA | Kappa | OA | AA | Kappa |
| Without intermediate state branch | 94.02 | 94.43 | 93.51 | 96.03 | 95.10 | 94.73 | 90.71 | 92.46 | 87.94 |
| With intermediate state branch | **96.61** | **96.53** | **96.34** | **99.11** | **98.50** | **98.81** | **92.90** | **93.33** | **90.68** |

(2) Ablation study for three-branch feature interaction (TBFI) modules: As a crucial component of the overall framework, the three-branch feature interaction module facilitates the interaction between hyperspectral and LiDAR branches by means of intermediate state branches, thereby enabling the further exploration and integration of their shared-complementary features. Through the experimentation of hyperparameter settings, we have discovered that cascading two TBFI modules together can effectively integrate advanced features from different modalities. Table 8 illustrates the classification performance of models before and after incorporating two TBFI modules into three RS datasets. After the incorporation of two TBFI modules, a significant improvement in classification performance is observed. On the Trento data set, OA is enhanced by up to 8.05%, which serves as compelling evidence that TBFI modules are highly effective in enhancing classification accuracy.

**Table 8.** OA, AA and kappa coefficients of the model before and after adding the TBFI modules in the three RS datasets.

| Dataset | Houston | | | Trento | | | MUUFL | | |
|---|---|---|---|---|---|---|---|---|---|
| | **OA** | **AA** | **Kappa** | **OA** | **AA** | **Kappa** | **OA** | **AA** | **Kappa** |
| Without TBFI modules | 93.98 | 93.47 | 92.94 | 91.06 | 92.40 | 90.01 | 90.06 | 91.84 | 86.42 |
| With TBFI modules | **96.61** | **96.53** | **96.34** | **99.11** | **98.50** | **98.81** | **92.90** | **93.33** | **90.68** |

(3) Ablation study for dynamic weight selection module: The dynamic weight selection module is designed to achieve a balanced representation of information across different modalities. It enables the adaptive assignment of weights to the output from three branches, followed by decision fusion, thereby addressing the issue of imbalanced data expression across various sensors. Table 9 shows the classification performance of the models before and after the introduction of the dynamic weight selection strategy on the three RS data sets. It can be seen that the classification performance of the model is improved to some extent after the introduction of the dynamic weight selection strategy, which proves that the DWS module can improve the classification accuracy of the model.

**Table 9.** OA, AA and kappa coefficients of models before and after the strategy were selected by dynamic weights on three RS datasets.

| Dataset | Houston | | | Trento | | | MUUFL | | |
|---|---|---|---|---|---|---|---|---|---|
| | **OA** | **AA** | **Kappa** | **OA** | **AA** | **Kappa** | **OA** | **AA** | **Kappa** |
| Without DWS module | 95.04 | 95.01 | 94.64 | 97.12 | 97.28 | 97.02 | 90.95 | 92.07 | 87.45 |
| With DWS module | **96.61** | **96.53** | **96.34** | **99.11** | **98.50** | **98.81** | **92.90** | **93.33** | **90.68** |

## 5. Conclusions

In this paper, a novel feature-decision level collaborative fusion network was proposed for remote sensing classification with hyperspectral and LiDAR data. The proposed approach seamlessly integrates feature-level and decision-level fusion strategies within a unified framework, effectively harnessing the combined advantages of both techniques. Specifically, at the feature level, we propose a novel three-branch feature interaction module to alleviate modality discrepancies. This enables us to fully leverage the shared and complementary features of hyperspectral and LiDAR data while achieving interaction and integration through intermediate state branches. A novel dynamic weight selection module is proposed at the decision level, which adaptively assigns weights to the outputs of three branches and integrates information from different modalities in a more balanced manner. Experimental results on three common remote sensing datasets fully demonstrate the effectiveness of our proposed framework.

The proposed three-branch feature interaction module and dynamic weight selection module offer a valuable reference for the fusion and classification of multi-modal

data in various fields. The proposed multilevel interactive fusion module fully performs feature interactive fusion of two different modal data through the middle branch, thus improving the classification accuracy. Furthermore, we plan to extend these modules to other domains, such as hyperspectral and multispectral fusion, as well as multispectral and panchromatic image fusion. Meanwhile, we will investigate methods to enhance the model's generalizability and its application in sensor data fusion across diverse modalities.

**Author Contributions:** Conceptualization, S.Z., Q.L. and X.M.; methodology, S.Z. and X.M.; software, S.Z.; validation, S.Z.; writing—original draft preparation, S.Z.; writing—review and editing, S.Z., Q.L. and X.M.; supervision, Q.L., X.M., G.Y. and W.S. All authors have read and agreed to the published version of the manuscript.

**Funding:** This research was funded by the National Natural Science Foundation of China (42171326); National Natural Science Foundation of China (42071323); Zhejiang Provincial Natural Science Foundation of China (LR23D010001, LY22F010014); and Ningbo Natural Science Foundation under Grant 2022J076.

**Data Availability Statement:** The Houston dataset is available at: http://dase.grss-ieee.org/ (accessed on 5 July 2022). The Trento dataset can be obtained from [34]. The MUUFL dataset can be obtained from [35].

**Acknowledgments:** The authors wish to acknowledge the team led by Hang Renlong of Nanjing University of Information Science & Technology who provided the code of the Coupled CNN.

**Conflicts of Interest:** The authors declare no conflict of interest.

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
