# Peer review of "Feature-Decision Level Collaborative Fusion Network for Hyperspectral and LiDAR Classification"

_remotesensing, doi:10.3390/rs15174148_

Round 1
Reviewer 1 Report
The authors skillfully utilize the intermediate state branch to bridge the HSI and LiDAR streams, thereby reducing the modality gap while integrating shared and complementary features of HSI and LiDAR data, as well as balancing information representation at the decision level through dynamic weighting. Overall, the proposed method is interesting and impressive, and the contribution is enough to provide a valuable reference for the joint classification field of HSI and LiDAR. However, several minor issues require addressing in this manuscript:
1. Please add graphical meanings such as addition and multiplication and S in Figure 1-4.
2. In the ablation experiment, the abbreviation TBFIMs in line 499 is inconsistent with the previous abbreviation TBFI, please keep it consistent, and so is the abbreviation DWSM.
3. In the conclusion, line 530, the use of model is not standard, module should be used, model generally means the whole model, please correct it, and please check the problem in the full text.
4. The reference format lacks the corresponding volume page number, such as 38, please check whether all reference formats are complete.
5. The comparison method used should at least be detailed in the Introduction section.
6. The classification result graph of SVM in Figure 10 is not consistent with other comparison graphs, so please keep it consistent.
Minor editing of English language is required.
Reviewer 2 Report
This paper proposed a novel feature-decision level collaborative fusion network for hyperspectral and LiDAR classification. It is acceptable after revising the following concern:
1. Avoid using first person.
2. Avoid lumping references as in [x, y] and all other. Instead summarize the main contribution of each referenced paper in a separate sentence.
3. It would be better if the authors compare their algorithm with algorithms in 2022 or 2023.
4. How is the performance of the proposed algorithm compared with the following algorithms: Hyperspectral Image Classification Based on Multiscale Cross-Branch Response and Second-Order Channel Attention. IEEE Transactions on Geoscience and Remote Sensing.
5. The results of your comparative study should be discussed in-depth and with more insightful comments on the behavior of your algorithm on various case studies. Especially on some datasets, it seems that the advantages of some classes of datasets compared to the comparative algorithm are not significant, which can be analyzed in-depth from a theoretical perspective.
6. Does the parameter selection have theoretical support? The experiment does not seem to have a comprehensive analysis of why the selection in the text was made.
7. Make sure your conclusions reflect on the strengths and weaknesses of your work, how others in the field can benefit from it and thoroughly discus future work.
8. The format of references needs to be modified.
The writing of the paper needs to be further polished to make it publishable.
Reviewer 3 Report
HOW HEIGHT INFORMATION INTEGRATED WITH SPECTRAL INFORMATION SHOULD BE MADE MORE CLEAR.
HOW HEIGHT INFORMATION INTEGRATED WITH SPECTRAL INFORMATION SHOULD BE MADE MORE CLEAR.
Round 2
Reviewer 2 Report
Accept after minor revision (corrections to minor methodological errors and text editing)
Minor editing of English language required
